# Synthesis and Characterization of Calcium Silicate Nanoparticles Stabilized with Amino Acids

**DOI:** 10.3390/mi14020245

**Published:** 2023-01-18

**Authors:** Anastasiya A. Blinova, Abdurasul A. Karamirzoev, Asiyat R. Guseynova, David G. Maglakelidze, Tatiana A. Ilyaeva, Batradz A. Gusov, Avetis P. Meliksetyants, Mari M. Pirumian, Maxim A. Taravanov, Maxim A. Pirogov, Dmitriy S. Vakalov, Tatiana V. Bernyukevich, Alexey A. Gvozdenko, Andrey A. Nagdalian, Andrey V. Blinov

**Affiliations:** 1Department of Physics and Technology of Nanostructures and Materials, Physical and Technical Faculty, North Caucasus Federal University, 355017 Stavropol, Russia; 2Faculty of Dentistry, North Ossetian State Medical University, 362025 Vladikavkaz, Russia; 3Faculty of Dentistry, Derzhavin Tambov State University, 392008 Tambov, Russia; 4Faculty of Medicine, Stavropol State Medical University, 355017 Stavropol, Russia; 5Medical and Preventive Faculty, Rostov State Medical University, 344022 Rostov-on-Don, Russia; 6Moscow State University of Civil Engineering, 129337 Moscow, Russia; 7Laboratory of Food and Industrial Biotechnology, North Caucasus Federal University, 355017 Stavropol, Russia

**Keywords:** nanoparticles, calcium silicate, amino acids, scanning electron microscopy

## Abstract

This work presents the development of a method for the synthesis of calcium silicate nanoparticles stabilized with essential amino acids. CaSiO_3_ nanoparticles were obtained through chemical precipitation. In the first stage, the optimal calcium-containing precursor was determined. The samples were examined using scanning electron microscopy. It was found that Ca(CH_3_COO)_2_ was the optimal calcium-containing precursor. Then, the phase composition of calcium silicate was studied using X-ray phase analysis. The results showed the presence of high-intensity bands in the diffractogram, which characterized the phase of the nanosized CaSiO_3_—wollastonite. In the next stage, the influence of the type of amino acid on the microstructure of calcium silicate was studied. The amnio acids studied were valine, L-leucine, L-isoleucine, L-methionine, L-threonine, L-lysine, L-phenylalanine, and L-tryptophan. The analysis of the SEM micrographs showed that the addition of amino acids did not significantly affect the morphology of the CaSiO_3_ samples. The surface of the CaSiO_3_ samples, both without a stabilizer and with amino acids, was represented by irregularly shaped aggregates consisting of nanoparticles with a diameter of 50–400 nm. Further, in order to determine the optimal amino acid to use to stabilize nanoparticles, computerized quantum chemical modeling was carried out. Analysis of the data obtained showed that the most energetically favorable interaction was the CaSiO_3_–L-methionine configuration, where the interaction occurs through the amino group of the amino acid; the energy value of which was −2058.497 kcal/mol. To confirm the simulation results, the samples were examined using IR spectroscopy. An analysis of the results showed that the interaction of calcium silicate with L-methionine occurs via the formation of a bond through the NH_3_^+^ group of the amino acid.

## 1. Introduction

Damage to bone tissue is considered to be one of the most severe injuries, as it is accompanied by severe pain and requires a long period of healing of the damaged area of the bone itself [1,2,3]. It is known that bones are of various shapes and that they have a complex structure which contributes to their light weight and high strength. In addition, bones perform many functions in the human musculoskeletal system: they provide mechanical support for the body; they are the basis for the attachment of the muscle tissues of the body; they protect internal organs; and they provide metabolic functions (preservation of calcium, phosphorus, and other necessary vital elements). In particular, bone structures are natural nanocomposites which consist of organic compounds based on collagen, as well as inorganic substances such as hydroxyapatite [4,5]. In order to maintain the constancy of the elemental composition of the body, as well as to strengthen bone tissues, an urgent task is to create bioactive materials that will be of similar phase composition to the main mineral components of bone tissues [6,7,8]. In addition, various branches of medicine (in particular, surgery and implantology) are engaged in the search for solutions to such problems [9,10,11,12,13].

Since the surface of dental implants is in direct contact with vital hard and soft tissues, and is also exposed to chemical, mechanical, and biological media, the implants must have certain properties, such as biological, mechanical, and morphological compatibility [14,15,16,17]. In addition, an important role for researchers in modern implantology is not only in the creation of implants from new materials, but also in the creation of various coatings for them. Various bioactive substances can be used as such materials. In particular, these include colloidal forms of metal silicates, as they have the necessary properties as well as good biological compatibility [18,19,20,21,22]. One can distinguish calcium silicate in the crystalline phase—wollastonite, due to its optimal physicochemical and antibacterial properties, as well as its ability to bind bone tissues [23,24,25]. In addition, nanosized calcium silicate is excellent for filling teeth, acting as an effective biologically active material [26,27,28,29,30].

Calcium silicate is widely used in medicine as a cement for the preparation of a solution associated with pulp regeneration and repair of hard tooth tissues, or to increase the osseointegration of implants [31]. Wang et al. (2015) found that a coating composed of nanoscale calcium silicate was able to reduce the rate of degradation of the implant, promoted the attachment of cells and their proliferation, and also increased the expression of agiogenic factors [32]. The authors of [33] have shown that nanoscale calcium silicate improves the bioactivity and osseointegration of implants and increases their service life.

The addition of amino acids to the composition of this material makes it possible to accelerate the process of bone tissue regeneration [34]. Ratirotjanakul et al. (2019) declared that the addition of amino acids makes it possible to accelerate the biodegradation process of nanomaterials in the human body [35]. It is important to note that Kamali and Ghahremaninezhad (2019) found that the addition of amino acids and proteins makes it possible to change the nanostructure and nanomechanical properties of calcium silicate hydrate [36]. This expands the potential uses of calcium silicate in medicine and biotechnology.

Thus, the analysis of literature data shows the current demand and scientific interest in nanoscale calcium silicate modified with biopolymers, and therefore, the purpose of this work was the synthesis and study of nanosized calcium silicate stabilized with amino acids.

## 2. Materials and Methods

### 2.1. Materials

The following materials were used in the experiment: sodium silicate (JSC “LenReaktiv”, St. Petersburg, Russia), calcium chloride (“HIMMED”, Moscow, Russia), calcium acetate (OOO ORT “Khimreaktivy”, Yekaterinburg, Russia), calcium nitrate (LenReaktiv JSC, St. Petersburg, Russia), and various amino acids: L-valine, L-leucine, L-isoleucine, L-methionine, L-threonine, L-lysine, L-phenylalanine, and L-tryptophan (Sigma-Aldrich, St. Louis, MO, USA). All materials were purchased in the form of dry powders.

### 2.2. Method for the Synthesis of Calcium Silicate Nanoparticles

Synthesis of calcium silicate nanoparticles was carried out through chemical precipitation in an aqueous medium. Calcium chloride, calcium nitrate, and calcium acetate were used as calcium precursors, and sodium silicate was used as a precipitant. In the first stage, using an accurate weighing method employing PIONEER PX84 analytical scales (Ohaus Corporation, Parsippany, NJ, USA), solutions of sodium silicate and a calcium-containing precursor with an optimal concentration of 0.8 M were prepared. The optimal concentration of the solution was defined as the concentration allowing the most aggregatively stable particles in the liquid phase to be obtained, and was established during a series of preliminary experiments.

### 2.3. Method for the Synthesis of Calcium Silicate Nanoparticles Stabilized with Amino Acids

The synthesis of calcium silicate nanoparticles stabilized with amino acids was accomplished through chemical precipitation in an aqueous medium (Figure 1).

Calcium acetate was also used as a calcium precursor, and sodium silicate was used as a precipitant. Various amino acids were employed to act as a stabilizer: L-valine, L-leucine, L-isoleucine, L-methionine, L-threonine, L-lysine, L-phenylalanine, and L-tryptophan. First, solutions of sodium silicate and a calcium-containing precursor with a concentration of 0.8 M were prepared using the accurate weighing method. Then, 0.27 wt% of stabilizer was added. Next, a solution of a calcium-containing precursor was added to the system. The resulting solids were washed and centrifuged. Finally, the washed precipitates were dried in an oven at 80 °C and ground in a porcelain mortar until homogeneous.

### 2.4. Methods for Studying Nanoparticles

Micrographs of the calcium silicate nanoparticles were obtained using a MIRA3-LMH scanning electron microscope (Tescan, Brno, Czech Republic) equipped with an AZtecEnergy Standard/X-max 20 elemental composition determination system (Tescan, Brno, Czech Republic).

The temperature transformations of the samples were studied through differential scanning calorimetry and thermogravimetry using a NETZSCH STA 449 F5 Jupiter instrument (Germany) equipped with the software package “NETZSCH Proteous-Thermal Analysis”, v. 6.1.0.

The phase composition of the samples was studied through X-ray phase analysis using an Empyrean diffractometer (PANalytical, Almeo, Netherlands) series 2 (PANalytical B.V., Almelo, Netherlands). The measurement range was 2θ = 25–90° (λ = 1.54 Å).

Quantum chemical modeling of the stabilization of the calcium silicate nanoparticles with amino acids was carried out utilizing the QChem program and the IQmol molecular editor (Q-Chem, Pleasanton, CA, USA). The calculations were carried out using equipment at the data processing center (Schneider Electric) of the North Caucasian Federal University.

The following amino acids were considered as potential stabilizers: L-valine, L-histidine, L-isoleucine, L-lysine, L-methionine, L-threonine, and L-cysteine. The calculations for the total energy and other characteristics were carried out using the following parameters: calculation: Energy; method: HF; basis: 3-21G; convergence: 5; and force field: Chemical. In relation to the calculations, the interaction of the calcium silicate molecule with an amino acid molecule was studied, and this was considered to be through the amino group of the amino acids and the oxygen of calcium silicate [37,38].

To study the vibrations of bonds of functional groups, the samples were examined with infrared spectroscopy using an FSM-1201 IR spectrometer with Fourier transform.

## 3. Results and Discussion

First, the effect of the type of calcium-containing precursor on the structure of the calcium silicate nanoparticles was studied through scanning electron microscopy. The results are presented in Figure 2.

Analysis of Figure 2 showed that the nature of the calcium-containing precursor significantly affected the structure of the calcium silicate samples. Samples obtained from calcium acetate and calcium chloride consisted of irregularly shaped aggregates formed by particles with diameters of 50–400 nm. The use of calcium nitrate made it possible to obtain irregularly shaped aggregates consisting of cubic calcium silicate with sizes from 0.3 to 2 μm. Thus, it was found that the optimal calcium-containing precursor was calcium acetate.

Then, the temperature transformations of the calcium silicate nanoparticles obtained via calcium acetate were studied using differential scanning calorimetry and thermogravimetry. The results of the study are shown in Figure 3.

An analysis of the results showed that the DTA curve of the CaSiO_3_ sample obtained from calcium acetate had minima in the range from 450 to 600 °C, which corresponded to endothermic processes of removal of water molecules from hydrate shells, as well as from the lattices of calcium silicate crystalline hydrates. In the range from 675 to 1500 °C, phase transitions of CaSiO_3_ occur from low-temperature modifications (wollastonite) to high-temperature modifications (para-wollastonite and pseudo-wollastonite). Thus, when heated above 1125 °C, wollastonite reversibly transforms into pseudo-wollastonite [39,40].

Next, the phase composition of the calcium silicate nanoparticles obtained from calcium acetate was studied using X-ray phase analysis. The results are presented in Figure 4.

Analysis of the diffraction pattern of the calcium silicate obtained from calcium acetate showed that high-intensity bands were obtained in the sample at 2θ = 26.21, 29.38, and 49.92°, which corresponded to the crystalline phase of wollastonite. In contrast, the use of calcium chloride and calcium nitrate as calcium-containing precursors was accompanied by amorphization of the CaSiO_3_ structure. The results obtained were consistent with other works [41,42,43].

In the next stage of the work, a study was carried out to determine the effect of the type of amino acid on the microstructure of the calcium silicate nanoparticles. The samples were examined using SEM microscopy. The results are shown in Figure 5 and the Appendix A.

Analysis of the data presented in Figure 5 and the Appendix A showed that the addition of amino acids did not significantly affect the morphology of the calcium silicate samples. The surface of the CaSiO_3_ samples, both without a stabilizer and with amino acids, was represented by irregularly shaped aggregates consisting of nanoparticles with a diameter of 50–400 nm.

In the next stage, computer quantum chemical modeling was carried out in order to determine the optimal configuration of the interaction of calcium silicate with various amino acids. For each interaction model, the total energy, the distribution of the surface potential, and the energies of the highest and lowest unoccupied molecular orbitals (HOMO and LUMO, respectively) were determined. The data obtained are presented in Table 1. The modeling results are presented in Figure 6.

Analysis of the data showed that the energies of the CaSiO_3_−amino acid molecular systems were significantly lower than the energies of the individual amino acid molecules. This fact indicated an energetically favorable formation of a chemical bond between the amino acid molecule and calcium silicate. As a result of data analysis, it was found that the most energetically favorable system was the “CaSiO_3_–L–methionine” system, the energy value of which was −2058.497 kcal/mol. The interaction between the surface of the CaSiO_3_ nanoparticle and L-methionine was realized through the amino group attached to the C_2_ atom of methionine. To confirm the quantum chemical modeling data, the samples were examined using IR spectroscopy. The results of the study are presented in Figure 7.

Analysis of the IR spectrum of L-methionine showed that in the region from 500 to 1800 cm^−1^, the presence of bands of deformation vibrations of bonds was observed: at 530 and 625 cm^−1^—-CH bonds; in the region from 708 to 804 cm^−1^—skeletal vibration bonds -CH_2_; at 874 cm^−1^—vibrations of the -C-C- bond; at 874 cm^−1^—vibrations of the -SH bond; at 953 and 984 cm^−1^—symmetrical bending vibrations of the carboxyl group COO^−^; at 1025 cm^−1^—vibrations of the CH_2_ bond; and at 1072 cm^−1^—deformation pendulum vibrations of the ionized amino group NH_3_. The region of bands from 1121 to 1152 cm^−1^ corresponded to vibrations of the -CH bond; at 1273 cm^−1^ to the fan-like vibrations of the -CH_2_ bond; at 1352 cm^−1^ to the bending vibrations of the ionized amino group NH_3_^+^; at 1406 cm^−1^ to the symmetrical vibrations of the carboxyl group COO^–^; at 1449 cm^−1^ to vibrations of the -CH_2_ bond; and the region of bands from 1515 to 1656 cm^−1^ corresponded to the symmetrical vibrations of the ionized amino group NH_3_^+^. In the same area, from 500 to 1800 cm^−1^, in the IR spectrum of calcium silicate stabilized with L-methionine, there were bands that are characteristic of bending vibrations: at 665 cm^−1^—Ca-O bonds; and at 883, 848, and 876 cm^−1^—symmetric vibrations of the O-Si-O bond; the region from 972 to 1177 cm^−1^ corresponded to asymmetric vibrations of the O-Si-O bond; that at 1340 cm^−1^ to Si-O-Si bonds; that at 1415 cm^−1^ to -CH_3_ bonds, and that at 1645 cm^−1^ to vibrations of the Si-N bond.

In the IR spectrum of calcium silicate, the area from 500 to 1800 cm^−1^ showed the presence of bands of deformation vibrations of bonds: at 538 and 658 cm^−1^—Ca-O bonds; at 818 cm^−1^—symmetrical vibrations of the O-Si-O bond; the region from 929 to 1080 cm^−1^ corresponds to asymmetric vibrations of the O-Si-O bond; that at 1346 and 1412 cm^−1^ to the Si-O-Si bonds; and the region from 1564 to 1645 cm^−1^ characterizes vibrations of the Si-O bond [43,44,45].

As a result of the analysis of the IR spectra, it was found that in the spectrum for L-methionine in the region from 1515 to 1656 cm^−1^, there was a decrease in the intensity of the bands, which corresponded to the symmetrical vibrations of the ionized amino group NH_3_^+^. In the IR spectrum of calcium silicate in the region from 1564 to 1645 cm^−1^, a decrease in the intensity of the bands was also observed, which, in turn, characterized vibrations of the Si-O bond. 

Based on the data presented above, it can be concluded that the interaction of the amino acid L-methionine with the surface of a calcium silicate particle occurs when silicon binds to amino groups in the amino acid molecule. The results obtained are consistent with the data obtained from computer quantum chemical simulation.

## 4. Conclusions

This work describes a study which was carried out to determine the effect of a calcium-containing precursor on the structure of calcium silicate nanoparticles. It was found that the nature of the calcium-containing precursor significantly affects the structure of the calcium silicate samples. Samples obtained from calcium acetate and calcium chloride consisted of irregularly shaped aggregates formed from particles with diameters of 50–400 nm. Analysis of the data showed that the optimal precursor for obtaining nanosized CaSiO_3_ was calcium acetate. The phase composition and temperature transformations of the samples of calcium silicate nanoparticles obtained from Ca(CH_3_COO)_2_ were also studied. Analysis of the data obtained showed that the samples were in the nanoscale state and had a crystalline modification—wollastonite.

The influence of the amino acid type on the surface structure of the calcium silicate nanoparticles was also studied. The following amino acids were used: L-valine, L-leucine, L-isoleucine, L-methionine, L-threonine, L-lysine, L-phenylalanine, and L-tryptophan. Analysis of the data presented in Figure 5 and the Appendix A showed that the addition of amino acids did not significantly affect the morphology of the calcium silicate samples. The surface of CaSiO_3_ samples, both without a stabilizer and with amino acids, was represented by irregularly shaped aggregates consisting of nanoparticles with diameters of 50–400 nm.

In order to determine the optimal amino acid for use in stabilizing calcium silicate nanoparticles, computer quantum chemical modeling was carried out. An analysis of the simulation results showed that the most energetically favorable interaction was the “CaSiO_3_–L-methionine” configuration, the energy value of which was −2058.497 kcal/mol. It is noteworthy that the interaction between the surface of the CaSiO_3_ nanoparticle and L-methionine was found to occur through the amino group attached to the C_2_ atom of methionine. 

To confirm the data obtained, samples of calcium silicate nanoparticles stabilized with amino acids were examined using IR spectroscopy. The results showed that in the IR spectra of L-methionine there was a significant decrease in the intensity of the bands in the region characteristic of symmetric vibrations of the ionized amino group NH_3_^+^. Additionally, in the IR spectrum of calcium silicate in the region from 1564 to 1645 cm^−1^, a decrease in the intensity of the bands was also observed, which, in turn, characterized vibrations of the Si-O bond. It was found that the stabilization of calcium silicate by the amino acid L-methionine was accompanied by the formation of a bond between silicon and the amino group of L-methionine. The results obtained were consistent with the results of the computer quantum chemical simulation.

## Figures and Tables

**Figure 1 micromachines-14-00245-f001:**
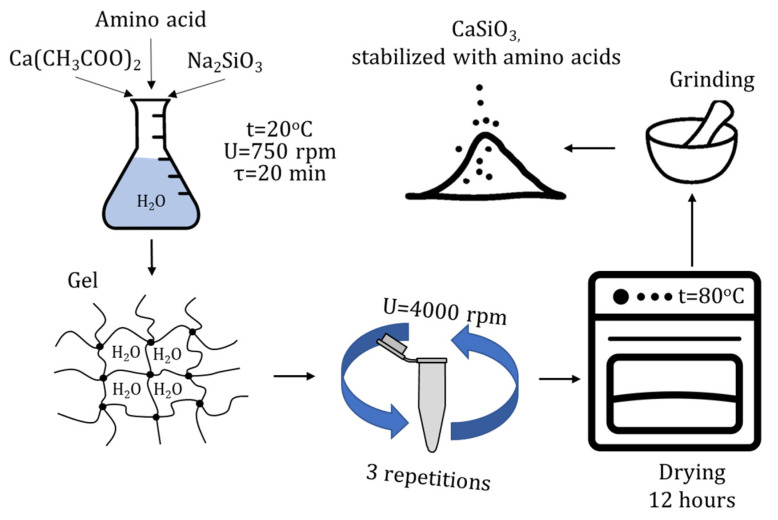
Scheme for the synthesis of calcium silicate nanoparticles stabilized with amino acids.

**Figure 2 micromachines-14-00245-f002:**
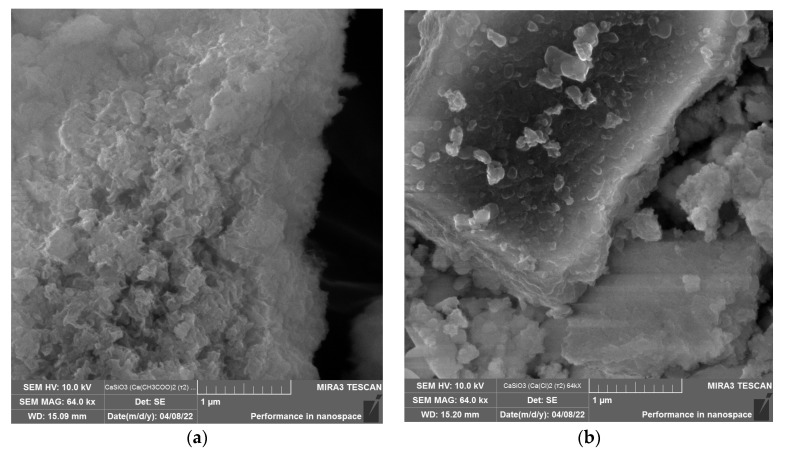
SEM micrographs of calcium silicate nanoparticles obtained from: (**a**) calcium acetate, (**b**) calcium chloride, (**c**) calcium chloride (other section), (**d**) calcium nitrate.

**Figure 3 micromachines-14-00245-f003:**
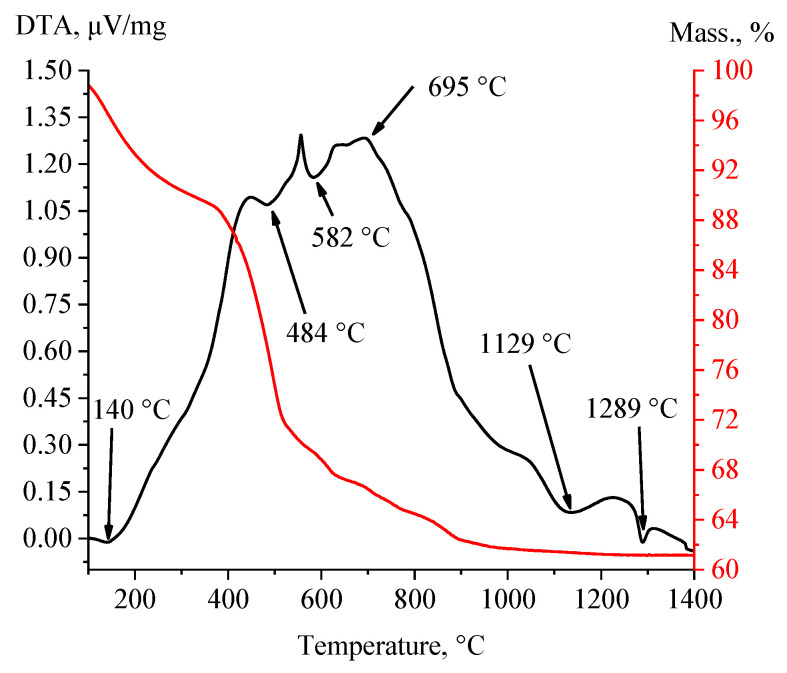
Derivatogram of calcium silicate obtained from calcium acetate.

**Figure 4 micromachines-14-00245-f004:**
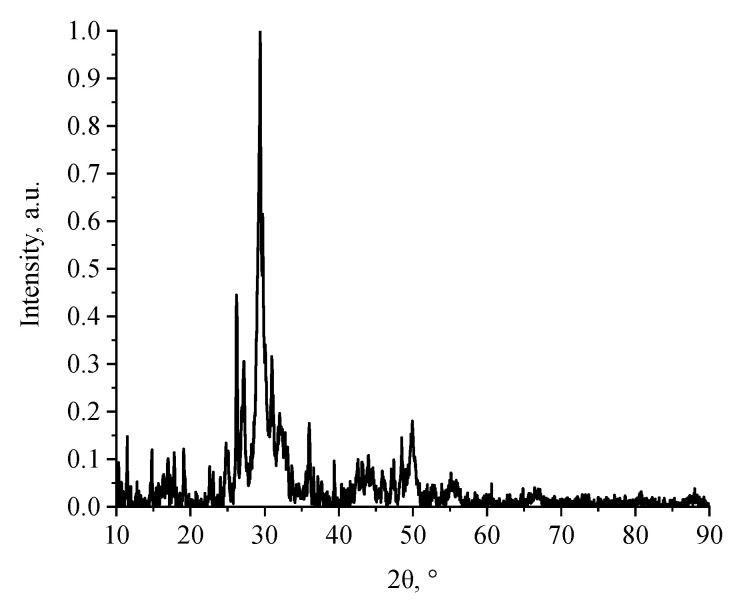
X-ray diffraction pattern of calcium silicate obtained from calcium acetate.

**Figure 5 micromachines-14-00245-f005:**
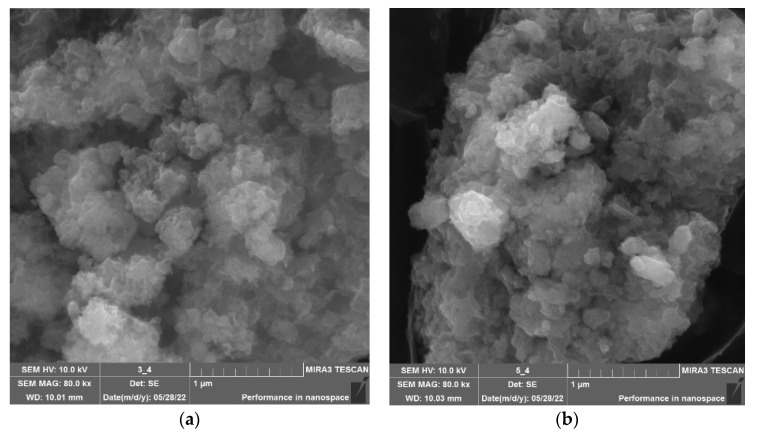
SEM micrographs of calcium silicate stabilized by: (**a**) L-phenylalanine, (**b**) L-leucine, (**c**) L-methionine, (**d**) L-threonine.

**Figure 6 micromachines-14-00245-f006:**
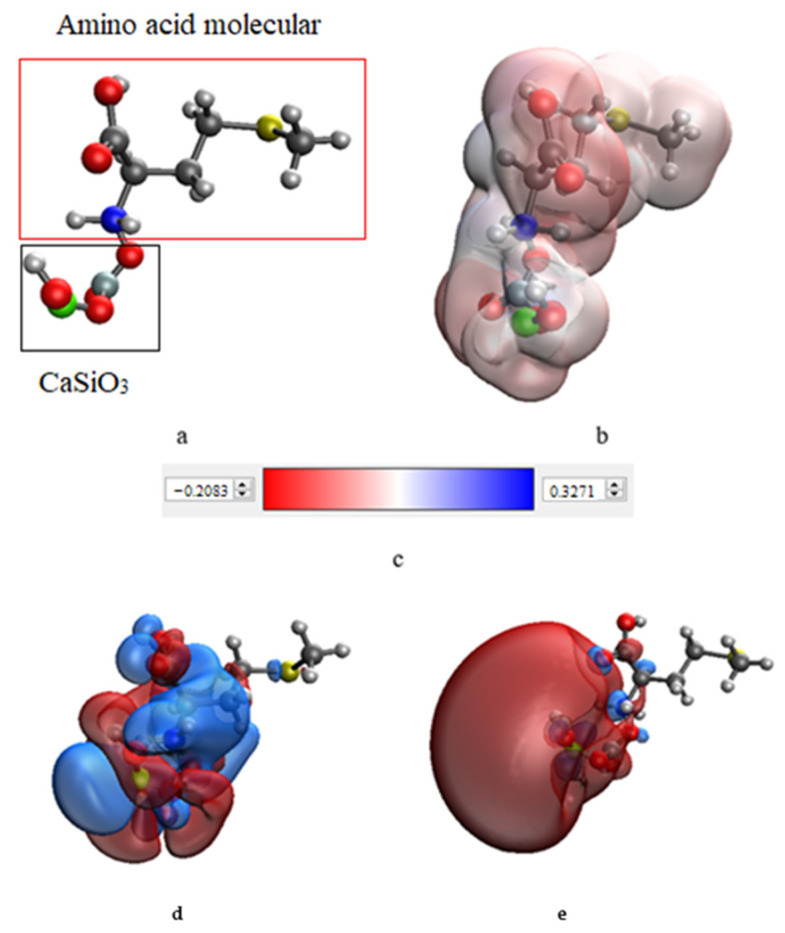
Modeling results for the CaSiO_3_–L–methionine molecular system, in which the interaction occurs through the amino group attached to the C_2_ atom of methionine: (**a**) model of the molecular complex, (**b**) electron density distribution, (**c**) electron density distribution gradient, (**d**) highest occupied molecular orbital, HOMO, (**e**) lowest unoccupied molecular orbital, LUMO.

**Figure 7 micromachines-14-00245-f007:**
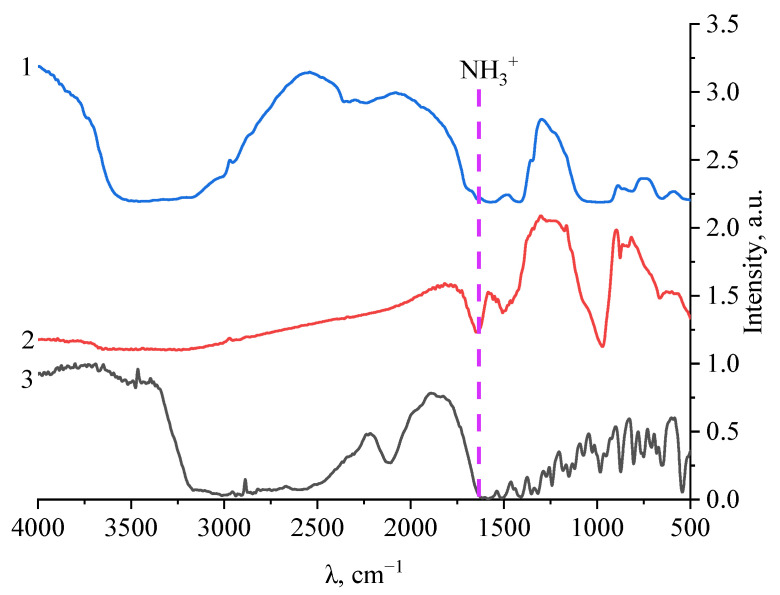
IR spectra: 1—calcium silicate; 2—calcium silicate stabilized with L–methionine; 3—L–methionine.

**Table 1 micromachines-14-00245-t001:** Modeling results of calcium silicate with amino acids.

Molecular System	Type of Interaction	E, kcal/mol	∆E, kcal/mol	E_HOMO_, eV	E_LUMO_, eV	η, eV
**CaSiO_3_**	-	−1185.674	-	−0.268	0.033	0.151
**CaSiO_3_–L-valine**	Amino acid	−402.112	-	−0.249	0.016	0.133
	Through the amino group attached to the C_2_ atom of valine	−1661.171	1259.059	−0.171	0.045	0.108
**CaSiO_3_–L-leucine**	Amino acid	−441.397	-	−0.260	0.006	0.133
	Via an amino group attached to the C_2_ atom of leucine	−1700.028	1258.631	−0.191	0.029	0.110
**CaSiO_3_–L-isoleucine**	Amino acid	−441.394	-	−0.247	0.018	0.133
	Via an amino group attached to the C_2_ atom of isoleucine	−1700.108	1258.714	−0.165	0.029	0.097
**CaSiO_3_–L-methionine**	Amino acid	−800.251	-	−0.232	0.006	0.119
	Via an amino group attached to the C_2_ atom of methionine	−2058.497	1258.246	−0.189	0.033	0.111
**CaSiO_3_–L-threonine**	Amino acid	−438.015	-	−0.248	0.006	0.127
	Through an amino group attached to the C_2_ atom of threonine	−1696.802	1258.787	−0.168	0.029	0.099
**CaSiO_3_–L-lysine**	Amino acid	−496.481	-	−0.177	−0.024	0.077
	Via an amino group attached to the C_2_ atom of lysine	−1754.871	1258.39	−0.140	0.043	0.092
	Via an amino group attached to the C_6_ atom of lysine	−1755.223	1258.742	−0.138	0.042	0.090
**CaSiO_3_–L-phenylalanine**	Amino acid	−554.424	-	−0.240	0.002	0.121
	Through an amino group attached to the C_2_ atom of phenylalanine	−1812.504	1258.08	−0.156	0.037	0.097
**CaSiO_3_–L-tryptophan**	Amino acid	−685.684	-	−0.195	−0.035	0.080
	Through an amino group attached to the C_2_ atom of tryptophan	−1943.146	1257.462	−0.162	0.035	0.099
	Through the secondary amino group of indole in tryptophan	−1942.821	1257.137	−0.168	0.021	0.095

## Data Availability

All data are available upon request from corresponding author.

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
