# Peer review of "Synthesis and Characterization of Calcium Silicate Nanoparticles Stabilized with Amino Acids"

_micromachines, 2023, doi:10.3390/mi14020245_

Round 1
Reviewer 1 Report
The paper presents experimental data on synthesis and characterization of calcium silicate nanoparticles stabilized with aminoacids. The paper describes a set of experiments, however representation of the work performed and results obtained is quite poor and requires substantial improvement. In general the paper critically lacks discussions in all the sections that makes difficult to evaluate the significance and reliability of the work done. Therefore I was hesitating between two recommendations: reject or major revisions. And I’ve finally preferred major revisions to give authors a chance to get published if they would be able to provide the necessary improvement.
One of my major concerns is that the main goal of the research is unclear, no explanation is given on what the particles are aimed to be used for, why their size in the nano domain is important for the intended applications, why it is necessary to stabilize them and how this stabilization was expected to be achieved by conjugation with aminoacids. And finally, nanoparticles were supposed to be synthesized, but in reality they came out not as individual particles but as extended agglomerates. How that corresponds to the paper title?
More specific commentaries and concerns are listed below.
The Introduction section should be extended considerably to include potential applications of the developed particles and state-of-the-art of the corresponding research worldwide.
Materials and methods section.
- In what form the materials were purchased.
- What was the “accurate weighing method”? Please specify the device used.
- Explanation of the workflow should be added to Fig. 1.
- Why it was necessary to stabilize the calcium silicate particles? Why it was performed using aminoacids and why the specific aminoacids were chosen?
- Self-referencing to papers 31,32,33 in sec. 2.4 does not seem appropriate. The authors use the standard equipment and referencing to own papers on different research objects can be done if some know-how developed in that research was applied in the current study. However none of the details is given.
- Please also describe the calculation procedure in more details.
Results section.
- Images from SEM in Figs. 2 and 5 are of quite low resolution. Even the legend below the images is blurred.
- I don’t see a lamellar structure in Fig. 2a, it rather can be seen in Fig. 2b.
- In general images present substantial agglomerates of different shapes rather than nanoparticles. As it is well known properties of individual particles and their agglomerates can be crucially different. In this sense there is a considerable discrepancy between the stated goal “synthesis of nanoparticles” and the result achieved…
- Fig. 3. Please explain the origin of the indicated minima in the derivatogram. Basing on what data the conclusion is made that “above 1125 °C wollastonite reversibly transforms into pseudo-wollastonite”? Please also explain data shown by the red curve.
- Fig. 6. Would be more pictorial in Fig. (a) if parts of the molecular system corresponding to CaSiO3 and methionine would be designated.
Some minor misprints.
- Line 136. Fig. 2 rather than Fig. 1.
- Line 193 calcium silicate rather than copper silicate.
- Fig. 7. Axis X. Please replace wavelength lambda by the wavenumber.
Author Response
We are grateful to the Reviewer 1 for his/her evaluation and for the time devoted to review our manuscript. All comments were useful and pleased us with the high level of understanding of the topic. We have addressed all recommendations as requested. All changes in the manuscript are marked by green. Please see the point-by-point answers below
The paper presents experimental data on synthesis and characterization of calcium silicate nanoparticles stabilized with aminoacids. The paper describes a set of experiments, however representation of the work performed and results obtained is quite poor and requires substantial improvement. In general the paper critically lacks discussions in all the sections that makes difficult to evaluate the significance and reliability of the work done. Therefore I was hesitating between two recommendations: reject or major revisions. And I’ve finally preferred major revisions to give authors a chance to get published if they would be able to provide the necessary improvement.
Thank you very much for your position and a chance to improve our work!
One of my major concerns is that the main goal of the research is unclear, no explanation is given on what the particles are aimed to be used for, why their size in the nano domain is important for the intended applications, why it is necessary to stabilize them and how this stabilization was expected to be achieved by conjugation with aminoacids. And finally, nanoparticles were supposed to be synthesized, but in reality they came out not as individual particles but as extended agglomerates. How that corresponds to the paper title?
Thank you very much for your comments. The aim of the series of works, which begins with this manuscript, is to create a coating for implants based on calcium silicate nanoparticles stabilized with amino acids, which will increase their biocompatibility. According to literature sources [1-3], the use of nanoparticles allows to reduce the rate of degradation of the implant, promotes the attachment of cells and their proliferation, and also increases the expression of agiogenic factors, as well as improves the bioactivity and osseointegration of implants and increase their service life. The addition of amino acids to the composition of calcium silicate makes it possible to accelerate the process of bone tissue regeneration [4], to accelerate the process of biodegradation of nanomaterials in the human body [5], to change the nanostructure and nanomechanical properties of calcium silicate hydrate [6]. The process of stabilization of calcium silicate nanoparticles by amino acids is considered in the text of the manuscripts. As a result of the synthesis, nanoparticles were obtained in the liquid phase. To study the samples by scanning electron microscopy, they were dried, then crushed. During drying, the nanoparticles were aggregated. In this regard, we see aggregates on micrographs. A micrograph has been added to the text of the article, which proves the formation of nanoparticles.
1 Wang X. et al. Fabrication of nano-structured calcium silicate coatings with enhanced stability, bioactivity and osteogenic and angiogenic activity //Colloids and Surfaces B: Biointerfaces. – 2015. – Т. 126. – С. 358-366.
2 Buga C. et al. Electrosprayed calcium silicate nanoparticle-coated titanium implant with improved antibacterial activity and osteogenesis //Colloids and Surfaces B: Biointerfaces. – 2021. – Т. 202. – С. 111699.
3 Ma R. et al. Osseointegration of nanohydroxyapatite-or nano-calcium silicate-incorporated polyetheretherketone bioactive composites in vivo //International journal of nanomedicine. – 2016. – Т. 11. – С. 6023.
4 Ozaki, K., Yamada, T., Horie, T., Ishizaki, A., Hiraiwa, M., Iezaki, T., ... & Hinoi, E. (2019). The L-type amino acid transporter LAT1 inhibits osteoclastogenesis and maintains bone homeostasis through the mTORC1 pathway. Science signaling, 12(589), eaaw3921.
5 Ratirotjanakul, W., Suteewong, T., Polpanich, D., & Tangboriboonrat, P. (2019). Amino acid as a biodegradation accelerator of mesoporous silica nanoparticles. Microporous and Mesoporous Materials, 282, 243-251.
6 Kamali, M., & Ghahremaninezhad, A. (2018). Effect of biomolecules on the nanostructure and nanomechanical property of calcium-silicate-hydrate. Scientific reports, 8(1), 1-16.
More specific commentaries and concerns are listed below.
The Introduction section should be extended considerably to include potential applications of the developed particles and state-of-the-art of the corresponding research worldwide.
Thank you for recommendation. We modified Introduction section.
Materials and methods section.
- In what form the materials were purchased.
Thank you for the question. All materials were purchased in the form of dry powders
- What was the “accurate weighing method”? Please specify the device used.
Thank you for the note. The method of accurate weighing involves weighing on analytical scales. Model and details of the analytical weights are indicated in the text of the article
- Explanation of the workflow should be added to Fig. 1.
Thank you for your comment. An explanation of the process shown in Figure 1 is provided in the following paragraph
- Why it was necessary to stabilize the calcium silicate particles? Why it was performed using aminoacids and why the specific aminoacids were chosen?
Thank you for your comment. Indeed, we missed explanation of this issue. We added this part in modified Introduction.
- Self-referencing to papers 31,32,33 in sec. 2.4 does not seem appropriate. The authors use the standard equipment and referencing to own papers on different research objects can be done if some know-how developed in that research was applied in the current study. However none of the details is given.
Thank you for the note. We added these references to show the readers application of mentioned methods in more details in our previous works. However, you are right, there is no innovative suggestions or know-how, so we removed them. Thank you again for your attentiveness.
- Please also describe the calculation procedure in more details.
Thank you for your comment. Calculation procedure was added to the manuscript.
Results section.
- Images from SEM in Figs. 2 and 5 are of quite low resolution. Even the legend below the images is blurred.
Thank you for the note. The quality of Figure 2 was improved. The data presented in Figure 5 were divided. Now there are only 4 the most important micrographs presented in figure 5. The rest micrographs were moved to Supplementary.
- I don’t see a lamellar structure in Fig. 2a, it rather can be seen in Fig. 2b.
Thank you for the note and your attentiveness. We corrected this issue.
- In general images present substantial agglomerates of different shapes rather than nanoparticles. As it is well known properties of individual particles and their agglomerates can be crucially different. In this sense there is a considerable discrepancy between the stated goal “synthesis of nanoparticles” and the result achieved…
Thank you for your comment. This occasion can be explained by the following. As a result of the synthesis, nanoparticles were obtained in the liquid phase. To study the samples by scanning electron microscopy, they were dried, then crushed. During drying, the nanoparticles were aggregated. In this regard, in microphotographs we see aggregates of nanoparticles. Nevertheless, we have added SEM micrographs, which show the presence of nanoparticles (50-400 nm).
- Fig. 3. Please explain the origin of the indicated minima in the derivatogram. Basing on what data the conclusion is made that “above 1125 °C wollastonite reversibly transforms into pseudo-wollastonite”? Please also explain data shown by the red curve.
Thank you for the comment. The explanation of the minima on the derivatogram is presented in the text of the publication. The conclusion that " above 1125 °C wollastonite reversibly transforms into pseudo-wollastonite" is associated with materials presented in the work Richet, P.; Robie, R.A.; Hemingway, B.S. Thermodynamic Properties of Wollastonite, Pseudowollastonite and CaSiO3 Glass and Liquid. Eur. J. Mineral. 1991, 3, 475-484, doi:10.1127/ejm/3/3/0475. The red curve describes the change in the mass of the sample. This curve is necessary in order to distinguish phase transitions from changes associated with the evaporation of chemically and physically bound water and other reaction products. The conclusions obtained in our work are based on the analysis of both curves.
- Fig. 6. Would be more pictorial in Fig. (a) if parts of the molecular system corresponding to CaSiO3 and methionine would be designated.
Thank you very much for recommendation. We agree and corrected it.
Some minor misprints.
- Line 136. Fig. 2 rather than Fig. 1.
Thank you, corrected!
- Line 193 calcium silicate rather than copper silicate.
Thank you, corrected!
- Fig. 7. Axis X. Please replace wavelength lambda by the wavenumber.
Thank you, corrected!
In addition, we gave our manuscript to native English speaker to improve language quality.
Reviewer 2 Report
The article “Synthesis And Characterization Of Calcium Silicate Nanoparticles Stabilized With Amino Acids” is related to nanoparticle synthesis and characterization, and as well as computer quantum chemical modeling of calcium silicate and various amino acids interaction. Unfortunately, the part related to synthesis and characterization is quite weak and requires significant improvement.
Scheme for the synthesis (Figure 1) contains step “Grinding” but in the "Materials and Methods" section this procedure isn't explained.
Why the 0.8M salt concentration was used in the particle synthesis?
SEM-images of nanoparticles (Figures 2 and 5) have low quality: scale bares are non-readable, and the morphology of particles is unclear.
The quality of SEM images should be improved if possible. Also, the the particle size distribution should be done on the basis of SEM images of particles. Also “nanoparticles” are not clearly be identified: there are aggregates of particles while single particles are barely distinguished, and lamellar structure of these single particles is not clear. Example of clear SEM-images of calcium silicate was presented in reference [https://doi.org/10.1186/s12951-016-0224-7].
The term “crystallites” (line 139) can’t be used in the case of particles size characterization. Crystallite size can be determined by XRD not by SEM.
Figures 2c and 5i demonstrate the same sample but they are look quite different.
X-ray diffraction pattern (Figure 4) is claimed to have amorphous structure, but there is no amorphous phase can be seen in the pattern.
The “Discussion” section should be added to the manuscript.
The English style should be improved.
Author Response
We are grateful to the Reviewer 2 for his/her evaluation and for the time devoted to review our manuscript. All comments were useful and pleased us with the high level of understanding of the topic. We have addressed all recommendations as requested. All changes in the manuscript are marked by green. Please see the point-by-point answers below
The article “Synthesis And Characterization Of Calcium Silicate Nanoparticles Stabilized With Amino Acids” is related to nanoparticle synthesis and characterization, and as well as computer quantum chemical modeling of calcium silicate and various amino acids interaction. Unfortunately, the part related to synthesis and characterization is quite weak and requires significant improvement.
Scheme for the synthesis (Figure 1) contains step “Grinding” but in the "Materials and Methods" section this procedure isn't explained.
Sorry for this occasion. We added he corresponding sentence in Materials and Methods section.
Why the 0.8M salt concentration was used in the particle synthesis?
Thank you for the note. During a series of preliminary experiments, we found that 0.8M salt concentration was the optimal concertation giving the most aggregatively stable particles in the liquid phase. We added corresponding explanation to the text.
SEM-images of nanoparticles (Figures 2 and 5) have low quality: scale bares are non-readable, and the morphology of particles is unclear.
Thank you for the note. The quality of Figure 2 was improved. The data presented in Figure 5 were divided. Now there are only 4 the most important micrographs presented in figure 5. The rest micrographs were moved to Supplementary.
The quality of SEM images should be improved if possible. Also, the the particle size distribution should be done on the basis of SEM images of particles. Also “nanoparticles” are not clearly be identified: there are aggregates of particles while single particles are barely distinguished, and lamellar structure of these single particles is not clear. Example of clear SEM-images of calcium silicate was presented in reference [https://doi.org/10.1186/s12951-016-0224-7].
Thank you for your comment and useful recommendations. We have done our best to improve the quality of figures. Initially we planned to get a particle size distribution, but unfortunately, the only one available equipment with software is under long-term maintenance.
The occasion associated with aggregates on SEM-micrographs can be explained by the following. As a result of the synthesis, nanoparticles were obtained in the liquid phase. To study the samples by scanning electron microscopy, they were dried, then crushed. During drying, the nanoparticles were aggregated. In this regard, in microphotographs we see aggregates of nanoparticles.
Nevertheless, we have added SEM micrographs, which show the presence of nanoparticles (50-400 nm).
The term “crystallites” (line 139) can’t be used in the case of particles size characterization. Crystallite size can be determined by XRD not by SEM.
Thank you for the note. We corrected this issue.
Figures 2c and 5i demonstrate the same sample but they are look quite different.
Thank you for the note. It was corrected.
X-ray diffraction pattern (Figure 4) is claimed to have amorphous structure, but there is no amorphous phase can be seen in the pattern.
Sorry for this occasion. There was a mistake in the text. We fixed and corrected it. “Calcium acetate as a calcium-containing precursor” was replaced by “Calcium chloride and calcium nitrate as a calcium-containing precursors”.
The “Discussion” section should be added to the manuscript.
Thank you for your attentiveness. We added Discussion.
The English style should be improved.
Thank you for recommendation. We gave our manuscript to native English speaker to improve language quality
Round 2
Reviewer 1 Report
The authors addressed all of my concerns and the paper can be accepted in the current form.
Reviewer 2 Report
The article was improved, but for correct characterization of obtained particles the size distribution (according to SEM) should be added. The size distribution as a function size on counts of particles should be presented, not only as bars on SEM-images.
I suggest to address this point to improve the work, prior to acceptance and publication.